# Temperature Control for an Intra-Mirror Etalon in Interferometric Gravitational Wave Detector Fabry–Perot Cavities

**Jonathan Brooks [1,*]**, **Maddalena Mantovani [1]**, **Annalisa Allocca [2,†]**, **Julia Casanueva Diaz [1]**, **Vincenzo Dattilo [1]**, **Alain Masserot [3]** and **Paolo Ruggi [1]**

1   European Gravitational Observatory, 56021 Cascina, Italy; mantovan@ego-gw.it (M.M.); julia.casanueva@ego-gw.it (J.C.D.); vincenzo.dattilo@ego-gw.it (V.D.); paolo.ruggi@ego-gw.it (P.R.)
2   Istituto Nazionale di Fisica Nucleare (INFN), Sezione di Pisa, I-56127 Pisa, Italy; annalisa.allocca@unina.it
3   Laboratoire d'Annecy de Physique des Particules (LAPP), University Grenoble Alpes, Université Savoie Mont Blanc, CNRS/IN2P3, F-74941 Annecy, France; Alain.Masserot@lapp.in2p3.fr
*   Correspondence: jonathan.brooks@ego-gw.it
†   Current address: Università di Napoli, Dipartimento di Fisica, Complesso Universitario Monte S. Angelo, 80126 Napoli, NA, Italy.

**Abstract:** The sensitivity of interferometric gravitational wave detectors is optimized, in part, by balanced finesse in the long Fabry–Perot arm cavities. The input test mass mirrors of Advanced Virgo feature parallel faces, which creates an etalon within the substrate, adding variability in the total mirror reflectivity, in order to correct imbalanced finesse due to manufacturing tolerances. Temperature variations in mirror substrate change the optical path length primarily through varying the index of refraction and are tuned to correct for a finesse imbalance of up to 2.8% by a full etalon fringe of 0.257 K. A negative feedback control system was designed to control the mirror temperature by using an electrical resistive heating belt actuator for a heat transfer process modeled as a two-pole plant. A zero controller filter was designed which achieves temperature control within 2.3% of the etalon fringe and recovers to within 10% of the working point within 32 hours after a step input of one etalon fringe. A preliminary unlock condition control designed to compensate when the interferometer unlocks shows that the control remains stable even after a drastic change in the plant due to the absence of the laser heating. Further improvements to the control must also consider the full heat transfer mechanisms by using modern control state space models.

**Keywords:** etalon; gravitational waves; interferometer; negative feedback control; classical control

## 1. Introduction

Advanced Virgo (AdV) is a recycled Michelson interferometric gravitational wave detector [1]. It is a second generation detector, part of a worldwide network, which along with LIGO, allowed for the recent discoveries in gravitational wave research [2–4]. Detection occurs at near complete destructive interference at a working point known as the dark fringe, where the interferometer (ITF) is sensitive to the differential signals generated by the passage of a gravitational wave. The high sensitivity of AdV is due, in part, to the 3 km Fabry–Perot arm cavities which enhance the light field by a factor of ~290. However, the sensitivity decreases with any asymmetry in the arm cavities due to the coupling of common noises, such as laser frequency or power noise, to the detector output [5,6].

One such asymmetry is due to differences of the finesse of each of the arm cavities, north and west. The finesse asymmetry, $\Delta\mathcal{F}/\mathcal{F}$, was restricted to below 1% for optimal detector sensitivity [1] after the effect on ITF performance parameters were studied [7,8]. The metric for the overall ITF

performance is the binary neutron star (BNS) range which is defined as the distance at which a single detector could observe the coalescence of a pair of 1.4 solar mass neutron stars with a signal-to-noise ratio of 8 [9]. The asymmetry of finesse spoils this parameter, and thus a lower finesse asymmetry corresponds to a maximum in the BNS range. Finesse is a function of the input and end test mass mirror reflectivites, which are turn is dependent on the mirror anti-reflective (AR) and highly reflective (HR) coatings. The finesse asymmetry requirement specified a mirror coating tolerance which surpassed manufacturing capabilities at the time of construction. These limitations were avoided by including the etalon effect into the AdV arm cavity design, first implemented in the previous generation Virgo+ detector [10].

The spherical input test mass mirrors feature parallel faces which establish an optical resonator, or etalon, within the mirror substrate between the AR and HR coatings. Instead, the end test mass has a wedge opposite the HR coated side, eliminating the possibility of an etalon, and fixes the mirror properties to those of the manufactured coating. The resonance condition of the etalon cavity contributes to the overall input mirror reflectivity. By tuning the etalon resonance, the arm cavity finesse is tuned independent of the coating properties.

Punturo [11] describes how the optical path length (OPL) dependent etalon resonance condition is varied by the input mirror substrate temperature and the subsequent effect on the arm cavity finesse. Figure 1 shows a simplified schematic of the arm cavity with relevant parameters for the etalon effect. The finesse is calculated by [12]

$$\mathcal{F} \approx \frac{\pi\sqrt{r'_A r_B}}{1 - r'_A r_B}\,,\tag{1}$$

where $r_B$ is the reflectivity of the end mirror. The overall input mirror reflectivity, $r'_A$, combines the mirror coating property with the etalon effect and is calculated by [11]

$$r'_A(L_m) = -r_{HR} + \frac{t_{HR}^2 r_{AR} e^{-2ikL_m}}{1 - r_{AR} r_{HR} e^{-2ikL_m}}\,,\tag{2}$$

where $t$ is the transmissivity, $r$ is the reflectivity, $L_m$ is the etalon optical path length, and the subscripts represent the AR and HR coatings of the mirror. As shown in Equation (2), the input test mass reflectivity is a function of $L_m$ which determines the resonance condition of the etalon cavity. As the etalon cavity medium is fused silica, $SiO_2$, instead of a vacuum, the OPL is a function of the substrate temperature [13]:

$$L_m(T) = n(T)(1 + \alpha_T T)L_{m,0}\,.\tag{3}$$

where $L_{m,0}$ is the nominal geometric length of the substrate at room temperature, $n(T)$ is the temperature-dependent index of refraction, and $\alpha_T$ is the coefficient of thermal expansion. For this analysis, it is assumed that there are no spatial thermal gradients in the mirror. Differentiating Equation (3) reveals that changes in the OPL are due to two mechanisms, thermorefractive and thermoelastic:

$$\frac{dL_m}{dT} = \left[\frac{dn}{dT}(1 + \alpha_T T) + \alpha_T n(T)\right]L_{m,0}\,.\tag{4}$$

where, for fused silica [14,15] at room temperature ($T \approx 293K$), $n = 1.450$, $dn/dt = 9.6 \times 10^{-6}K^{-1}$ and $\alpha_T = 5.1 \times 10^{-6}K^{-1}$. The change in temperature required to change the OPL through a complete free spectral range, termed etalon fringe, is calculated by solving Equation (4) for $dT$ with $dL_m = \lambda/2 = 532$ nm for the infrared beam and $L_{m,0} = 0.2$ m.

$$\Delta T = 0.257K\,.\tag{5}$$

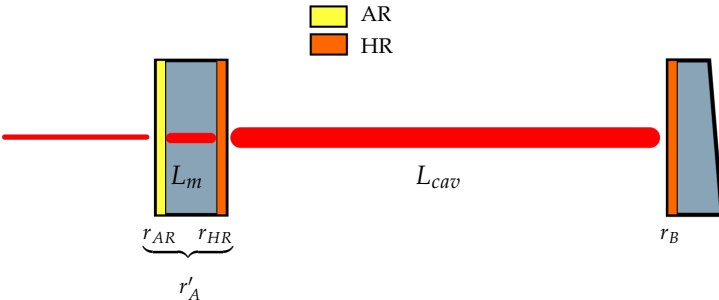

**Figure 1.** Schematic of a Fabry–Perot arm cavity with the etalon effect shown in the input test mass substrate and relevant parameters.

The effect of the arm cavity finesse throughout an etalon fringe was predicted by using a FINESSE [16] frequency domain simulation. The results of the simulation are shown in Figure 2. With the measured mirror coating parameters, the etalon effect is able to correct a finesse asymmetry up to 2.8%. These results also indicate that an uncontrolled etalon effect will push the finesse asymmetry out of the 1% requirement. The optimum working point occurs when the two cavity finesses are equalized at the maximum of the west arm finesse; the BNS range is maximized by minimizing the finesse asymmetry and maximizes the cavity finesse. Instead, in order to determine the temperature requirement, the worst case scenario (WCS) is considered, wherein the finesse is matched at the maximum slope of the derivative, $F = 449.2$, and the north and west test masses must maintain mirror temperature accuracies of 14 and 19 mK, respectively, in order to remain within the 1% finesse asymmetry requirement. This tolerance band is shown by the horizontal lines in Figure 2. These temperature tolerances agree well with measured BNS range fluctuations as a function of mirror temperatures [17]. Assuming that the working point corresponds with one of the peaks in the fitted sine functions, in order to maintain 1% BNS range tolerance, the north and west mirror temperature must maintain accuracies of 25 and 21 mK, respectively.

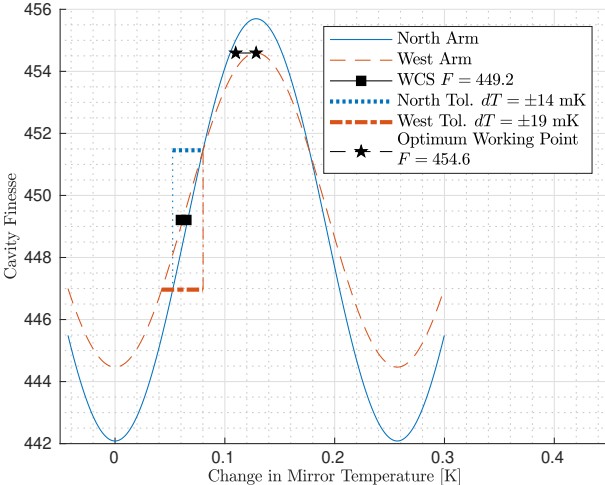

**Figure 2.** Finesse variation due to the etalon effect for the two arm cavities of Advanced Virgo. The temperature tolerance is calculated in the worst case scenario (WCS), where the slope of the curve is maximal. The reflectivity of the anti-reflective (AR) coating is 58ppm for the north arm and 32 ppm for the west arm. The reflectivity of the highly reflective (HR) coating, instead, is 0.98625 for the north arm and 0.98623 for the west.

Etalon control was implemented in Virgo on a previous generation of the detector, Virgo+ [18,19]. The gain of this controller was tuned to achieve an accuracy of 30 mK, which greatly stabilized the performance of the ITF, particularly in the BNS range. This control replaced the previous "human in the loop" method [20], where the correction voltage to the actuator was manually adjusted every few days. The physical configuration of the ITF changed appreciably from Virgo+ to AdV such that this control was no longer directly compatible. In particular, the etalon mirror volume increased significantly, which decreased the effective power of the actuator.

This paper presents the current application of the etalon control in AdV. The control system is developed in detail, including a development of the thermodynamic heat transfer system and justification for the two-pole plant model representation. Two controller filter designs are presented and compared against each other and the uncontrolled case in terms of temperature accuracy and recovery time after a step input. Finally, a separate unlock condition control methodology is developed to account for the drastic plant change when the ITF unlocks and the laser heating is removed.

## 2. Control System Description

Several assumptions are necessary to describe the non-linear physical system as the plant for a linear time-invariant (LTI) negative feedback control system. The temperature of the mirror is governed by the closed thermodynamic system defined by the physical boundaries of the mirror. Figure 3 shows a schematic of the thermodynamic system with relevant components and heat transfer labeled. The system is controlled by modulating the resistive heating belt actuator installed around the circumference of the vacuum tower located below the mirror [21–23]. The physical boundaries of the tower are treated as a second, coupled thermodynamic system to define the complete heat transfer from the heating belt actuator to the mirror temperature.

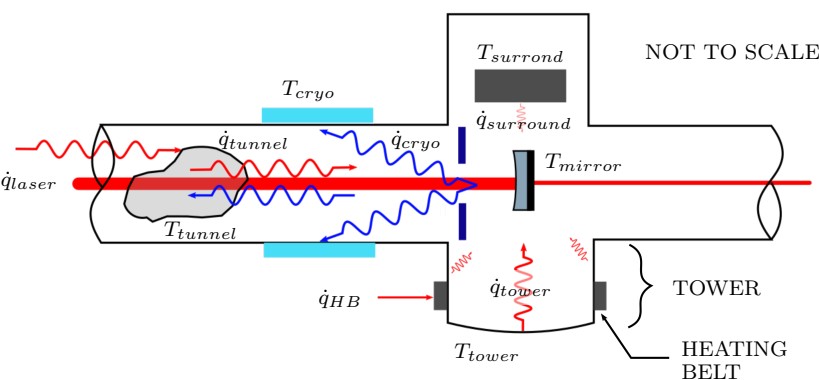

**Figure 3.** Identified radiative heat transfer sources relevant to the etalon control; also shown is the heat transfer due to the heating belt actuator, $\dot{q}_{HB}$. Red arrows indicate heat transfer into the mirror, blue indicate heat transfer out.

### 2.1. Physical Plant Description

The temperature of the mirror is determined by the first law of thermodynamics. No work is done on or by the system due to the low mirror velocity in the order of 1 $\mu$m/s, and very little in the way of elevation changes due to the pendulum motion. The primary mechanism for changes in internal energy is the change in internal temperature because of the low coefficient of thermal expansion for fused silica. The mirror is suspended in ultra high vacuum by fused silica suspension wires; therefore, conduction and convective heat transfer are negligible. Therefore, the first law of thermodynamics may be simplified to

$$c_v \frac{\mathrm{d}T_m}{\mathrm{d}t} = \sum_{i=1} \dot{q}_{\mathrm{rad},i}, \tag{6}$$

where $c_v$ is the specific heat at constant volume of the mirror and $T_m$ is the mirror temperature. The radiative heat transfer term, $\dot{q}_{\mathrm{rad},i}$, has contributions from laser irradiation and gray body radiation.

The heat transfer from the main laser beam resonating in the Fabry–Perot arm cavities is proportional to the power resonating in the cavities, $P$, and the absorption of the mirror at the laser wavelength $\alpha(\lambda)$:

$$\dot{q}_{\mathrm{r,laser}} \propto \alpha(\lambda)P, \tag{7}$$

This is a major contributor to the mirror steady state temperature. In fact, a separate etalon control configuration was designed to compensate for the lack of this heat transfer source when the ITF unlocks—discussed further in Section 3.3.

For gray body radiation [24], the heat transfer is proportional to the difference in temperatures to the 4th power and the view factor, i.e., how much radiation from one surface is "seen" by the mirror surface:

$$\dot{q}_{\mathrm{r},i} = \epsilon_i \sigma A_i F_{i \to m}(T_m^4 - T_i^4), \tag{8}$$

where $\sigma$ is the Stefan–Boltzmann constant, $\epsilon_i$ is the emissivity of surface $A_i$, and $F_{i \to m}$ is the view factor of surface $A_i$ to the mirror surface $A_m$.

The actuator is coupled to the mirror temperature by the tower temperature, $T_t$ which is substituted for $T_i$ in Equation (8). The tower temperature is also governed by the first law of thermodynamics with additional heat transfer terms for conduction from the heating belt actuator and convection to the exterior.

$$c_{v,t} \frac{\mathrm{d}T_t}{\mathrm{d}t} = \dot{q}_{HB} + h(T_t - T_\infty) + \sum_{j=1} \dot{q}_{\mathrm{rad},j} \tag{9}$$

where $c_{v,t}$ is the specific heat at constant volume of the tower, and $h$ is the convective heat transfer coefficient between the tower exterior surface and ambient air at temperature $T_\infty$. For this analysis, the tower is assumed to have no spatial thermal gradients so that the conduction heat transfer term is simply equal to the Joule heating of the heating belt actuator, $\dot{q}_{HB}$.

## 2.2. Plant Modeling

Two major assumptions are made in order to analyze the physical system as a LTI system. First, the inherently non-linear radiative heat transfer process is linearized [25] over the small changes in the temperature during etalon control by using the thermal resistive representation

$$\dot{q}_{\mathrm{r},i} = -\frac{T_m - T_i}{R_r}, \tag{10}$$

where the thermal resistance term, $R_r$ is given by

$$R_r = \left[4\epsilon\sigma T_i^3 A_i F_{i \to m}\right]^{-1}. \tag{11}$$

Second, with the exception of radiative heat transfer from the tower to the mirror, $\dot{q}_{tower}$, all other radiative heat transfer mechanisms are considered constant with respect to the control loop timescales. Therefore, the heat transfer contribution of these sources is treated collectively as a damping coefficient on the temperature changes in Equations (6) and (9). This includes the uncontrolled heat transfer sources of the surrounding structures, of the cryotrap, and from the arm tunnels, $\dot{q}_{surround}$, $\dot{q}_{cryo}$, and $\dot{q}_{tunnel}$, respectively. With this assumption, daily and seasonal fluctuations in temperature, $T_{\mathrm{tunnel}}$ and $T_{\mathrm{surround}}$, and systematic changes in laser power may be compensated for by the term $R_r$ as a constant gain on the plant.

To derive the system equation, one final simplification is necessary. To make Equations (6) and (9) linear, the temperatures $T_m$ and $T_t$ are replaced with the difference $\Delta T_{m/t} = T_{m/t} - T_\infty$ so that the convective term in (9) is linear. Finally, the system equation is derived by taking the Laplace transform of Equations (6) and (9) and combining to find

$$H(s) = \frac{\Delta T_m(s)}{\dot{q}_{HB}} = \frac{1}{c_v c_{v,t} R_r s^2 + [c_{v,t} - hR_r c_v + c_v] s - h} \tag{12}$$

which is a two-pole plant of the general form

$$H(s) = \frac{K}{(s + p_1)(s + p_2)}. \tag{13}$$

Due to the large amount of uncertainty in many of the measured parameters in Equation (12), the plant response as a whole was measured instead of the individual parameters.

### 2.3. Plant Measurement

The plant transfer function was identified by measuring the mirror temperature response to a step input of the heating belt actuator. Figure 4 shows the step response of the north input mirror temperature. The north plant is shown as representative of both the north and the west owing to the similarities in the two plants. As the step response is on the order of days, different steps of 10 and 6 V were applied to the north and west arm actuators, respectively, to maximize information gained about the plant. A double-pole system response was fit to the data with a coefficient of determination of $R^2 = 0.9998$ for poles at 38.2 and 5.26 $\mu$Hz. The west mirror had a slightly worse fit with $R^2 = 0.9125$. The higher frequency pole is attributed to the tower heating and is visible as the concave up portion at the beginning of the response. The slower, mirror heating pole is responsible for the longer response time to reach steady state. The zoomed portion of the plot shows a dip in the measured data compared to the modeled response centered at $1 \times 10^5$ seconds. This is a period where the ITF unlocked, meaning that the laser was no longer heating the mirror. These data were included in the fit due to the fact that the model parameter estimation method required evenly spaced time domain data. Table 1 lists the poles for the north and west plants.

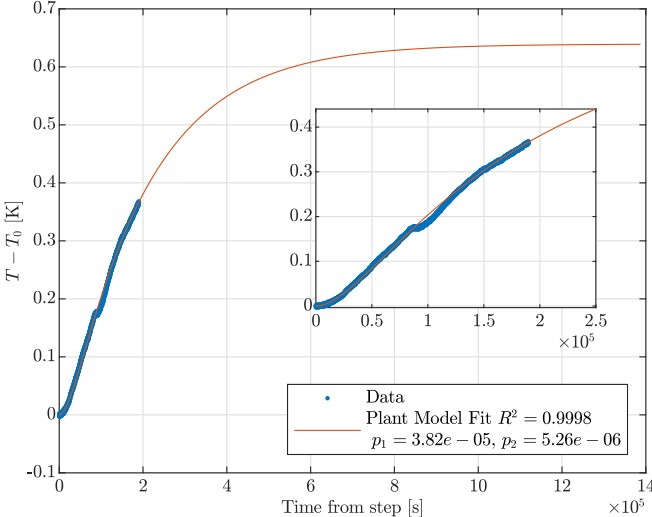

**Figure 4.** Two-pole plant model estimation for the north mirror step response. A system response with poles at 38.2 and 5.26 $\mu$Hz fit to the data yielded $R^2 = 0.9998$. The slight dip in the data is due to the ITF unlocking. The west arm exhibits similar behavior, albeit a slightly less goodness of fit.

**Table 1.** Plant properties for actuator and laser heating

| Plant | First Pole [Hz] | Second Pole [Hz] |
|---|---|---|
| North Plant | $3.82e{-}05$ | $5.26e{-}06$ |
| West Plant | $2.85e{-}05$ | $5.26e{-}06$ |

### 2.4. Control System Block Diagram

The etalon temperature control system may now be represented as a linear, time-invariant, negative feedback control system, as shown in Figure 5. The primary sensor is a thermistor attached to the temperature compensation system (TCS) ring heater [1] surrounding the mirror. A temperature sensor cannot be attached directly to the mirror itself because the presence of the sensor wire will detrimentally effect the suspension control and thermal noise performance [1]. The ring heater was powered off during the present analysis and the proximity of the ring heater to the mirror made this a reliable measurement of the mirror temperature. The ring heater cannot be used as the etalon actuator because it would cause detrimental thermal lensing [26]. If the ring heater is used for TCS in the future, the temperature may be measured by using the calibrated locations of temperature sensitive drum mode resonant frequencies [11,27–29]. Regardless of the sensor used, the signal is then calibrated to get the measured temperature of the mirror.

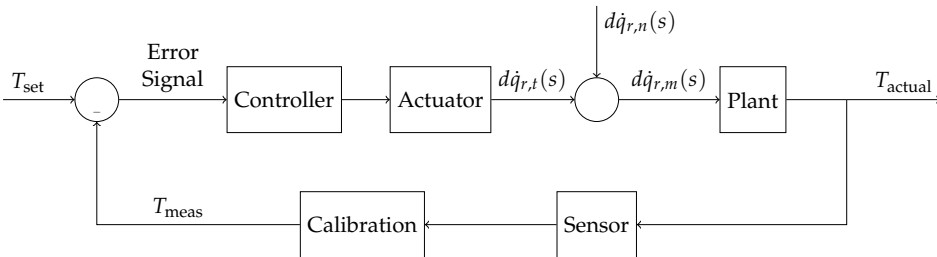

**Figure 5.** Etalon control system block diagram identifying main components and relevant variables.

The controller is the focus of this study. A correction signal is sent to the actuator, linearly dependent on the error signal. The error signal is the measured temperature subtracted from the desired set temperature. The set temperature is selected as that which maximizes the BNS range, as determined by previous measurements [17], and is cyclic, with a period of the fringe width (0.257 K). In this convention, a negative error signal corresponds to the mirror being hotter than the set temperature, and a positive error corresponds the mirror being colder than the set temperature.

Noise is included in the control system between the actuator and the plant. The noise represents heat transfer sources at off-nominal conditions or any unaccounted for heat transfer.

### 2.5. Controller Design

Two digital etalon temperature control filters were designed and implemented. The north control, described in detail, is to be representative of both arms, as the control filters are nearly identical. The first filter is an integrator with a gain, termed simple integrator (SI). The gain was tuned so that: (1) the unity gain frequency (UGF) of the open loop transfer function was below that of the plant pole frequency, necessary for a stable loop, and (2) the actuator would not saturate for a step input of one fringe. The primary goal of this control filter was to stabilize the performance of the ITF. At the time of the implementation, transients in the etalon disrupted the isolation and study of other shorter time constant noise sources present in the ITF by adding an unknown variable to long term studies. The speed to reach the desired accuracy was not as important as reducing transients in the mirror temperature.

The second filter, termed the zero controller, improves the speed and accuracy of the controller by adding a zero to the gain/integrator filter. Essentially, the filter will give a large kick to the actuator at the onset of a step instead of slowly building the correction as the SI filter does.

Figure 6 compares the simple integrator and zero controller open loop transfer function bode diagrams, and the closed loop mirror temperature response to an error step input of one etalon fringe (0.257 K). The time the zero controller took to reach within 10% of the steady state temperature was 6.4 times less than the SI controller did, as is evident in the difference in time scales between Figure 6b,c. The gain on the SI filter was rather conservative in order to avoid run-away temperatures exhibited during testing of the etalon control. The UGF was set approximately an order of magnitude below the lower pole frequency. The UGF of the zero controller was well above the lower pole frequency; however, it was stable due to the addition of the zero.

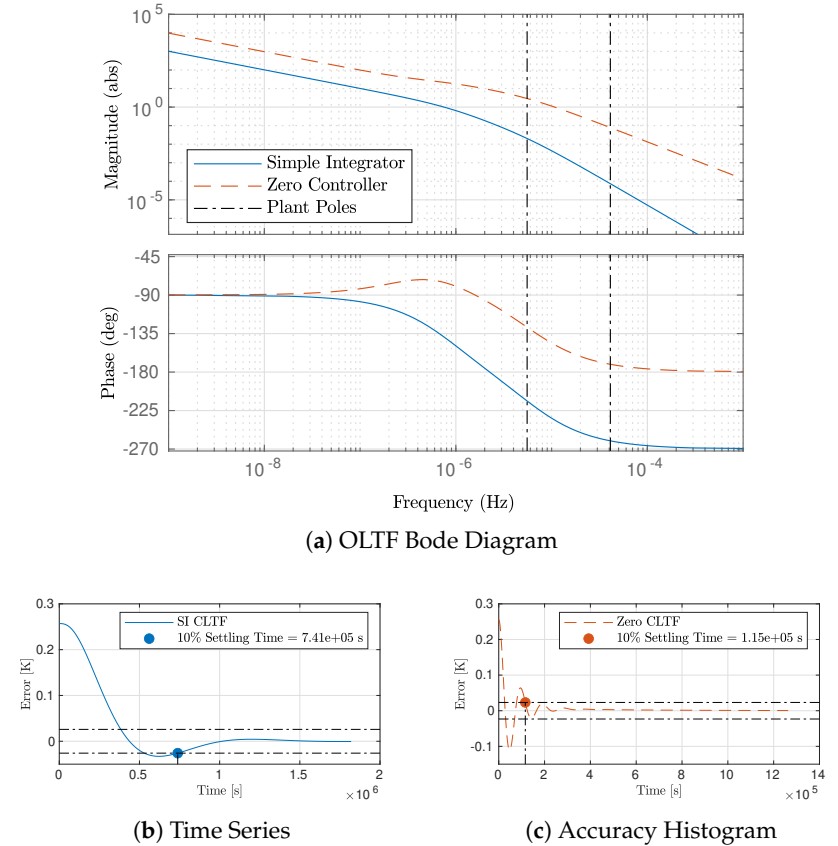

(**a**) OLTF Bode Diagram

(**b**) Time Series    (**c**) Accuracy Histogram

**Figure 6.** Comparisons of the simple integrator and zero controller via open loop transfer function bode diagram and closed loop mirror temperature response to an etalon fringe (0.257 K) step input to the error. The vertical lines in the Bode diagram indicate the two plant poles. Note the step responses are plotted on different time scales, since the settling times are separated by several orders of magnitudes.

## 3. Results

### 3.1. Recovery Speed after Unlock

Because of the large number of unlocks due to seismic activity, meteorological events, manual settings, and detector glitches, it is important for the etalon to recover quickly after an unlock. After an unlock, the error may be at an arbitrary value depending on the length of the unlock, tunnel temperature, error at unlock, etc. The speeds of the control filters are compared by finding the time it takes the temperature error to reach $1/e$ of the initial value after an unlock or a temperature set point change. The data for each of the control filters were chosen such that the temperature error

was sufficiently large and the ITF remained locked after the starting point for the temperature error to reduce to at least $1/e$. The available events for the zero controller were limited, as the temperature error was generally low, within 50 mK.

Figure 7 shows the speed comparison between the simple integrator and the zero controller. The time constant of the zero controller is five times faster than the simple integrator.

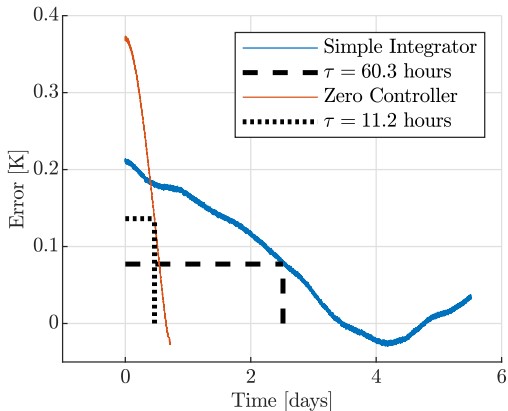

**Figure 7.** Controller speed after interferometer (ITF) unlock with initial temperature error. The zero controller is five times faster than the simple integrator.

## 3.2. Controller Accuracy

The controller accuracy was determined by calculating a probability density function (PDF) of the mirror temperature error for the longest lock period during each control filter implementation. In order to fully characterize the behavior, the accuracy of the uncontrolled mirror temperature was also analyzed. For the uncontrolled case, the error is defined as the difference between the closest optimal etalon working point that maximizes detector sensitivity and the measured temperature. The longest lock of all three cases was 132 h (5.5 days), which is the maximum possible due to the intentional unlocks for calibration held on Wednesday evenings and the maintenance period on Tuesday mornings.

Figure 8 compares the PDFs of the three cases. A normal distribution was fit to each by using the mean and standard deviation. The uncontrolled case exhibits poor accuracy at almost half fringe, albeit a small standard deviation. The behavior was expected, as the primary actuator on the mirror temperature during the long lock of the uncontrolled etalon consists of day/night tunnel temperature fluctuations. The accuracy is somewhat arbitrary and is highly seasonally dependent, even weekly dependent.

The best performance was given by the zero controller. The accuracy is remarkable—a mean error of 1 mK with a standard deviation of 6 mK. The temperature error was relatively low at the start of the lock period. However, given the speed of the controller presented in Section 3.1, the control filter is assumed to have similar performance with a larger initial error.

The simple integrator shows noticeably increased accuracy with a much lower standard deviation than the uncontrolled case. Note that the 0.2 K initial temperature error was significantly higher than that of the zero controller. The actuator driving the error to zero constituted the majority of the lock, which can be seen by the intermediate data points in Figure 8. It is difficult to properly characterize the performance of the simple integrator, as the time it takes to reach the set temperature is comparable to the maximum allowable lock period. However, given the speed of the simple integrator control and the number of unlocks exhibited by the ITF, this is representative of the anticipated long term performance.

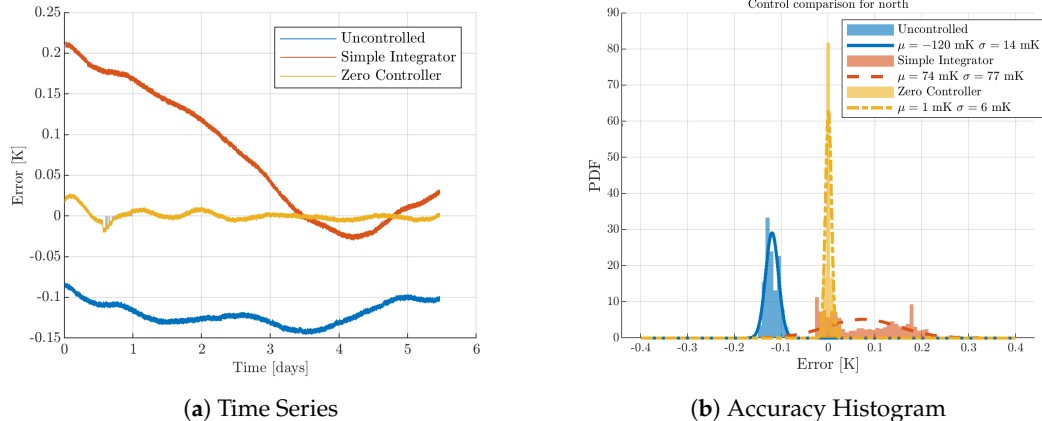

(**a**) Time Series

(**b**) Accuracy Histogram

**Figure 8.** Probability density function of mirror temperature error comparing the accuracy of the simple integrator, zero controller, and uncontrolled cases during a lock period of 5.5 days. The zero controller has remarkably higher accuracy.

### 3.3. Unlock Condition Control

The last design consideration for the etalon control is a methodology for when the ITF unlocks. After an unlock, the mirror rapidly cools without the laser heat transfer. Likewise, the mirror rapidly heats at the subsequent lock acquisition. The mirror temperature change due to the ITF unlock and lock acquisition represents a step response of a single-pole system. The system may be derived from Equation (6) focusing on the mirror temperature and the laser power:

$$c_v \frac{dT_m}{dt} = \alpha_\lambda \dot{q}_{r,laser} + \sum_{i=1} \dot{q}_{\text{rad},i}, \tag{14}$$

where the radiation heat transfer term on the right-hand side is proportional to the mirror temperature, $T_m$.

Figure 9 shows that the mirror cooling and heating after an unlock with the step response of a single-pole system fit to the data. The fit agrees very well with the data as the 95% confidence interval only has a small deviation from the fitted poles. The laser plant is approximately an order of magnitude faster than the actuator plant given in Table 1. The two laser plants are expected to be equivalent; however, there is a slight discrepancy in the measured poles. These measurements are considered preliminary and were used for the initial development of the unlock condition. Further measurements are necessary to explain this discrepancy.

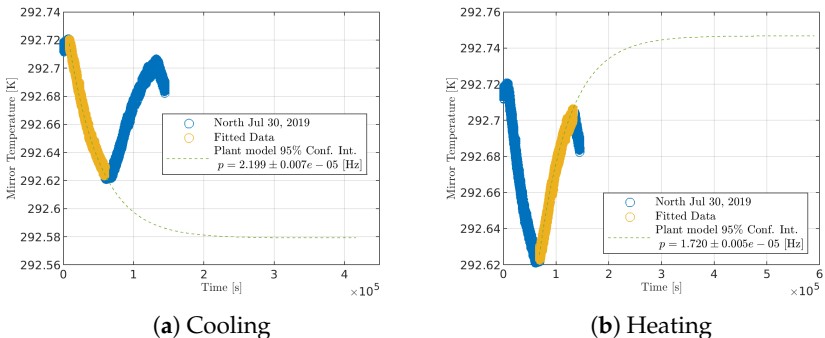

(**a**) Cooling

(**b**) Heating

**Figure 9.** Single-pole plant modeling for mirror temperature response to a step input of the laser power representing unlock and subsequent lock acquisition. The modeled plant was used to determine a error offset for the unlock condition control.

Due to the speed of the laser plant, the defined etalon control cannot be used during an unlock period. The actuator corrections will grow artificially large as the heating belt attempts to compensate for the laser heat transfer, leading to significant overshoot errors when the ITF relocks. Two methods have been employed to allow for etalon control during unlock periods. Initially, when the ITF unlocked, the last correction value was held constant until the ITF relocked. The value was filtered by a clipping function using a known, manually set steady state value to avoid an overly large or small correction. However, this method is not the ideal because the manually set steady state correction voltage is seasonally dependent and the error may continue to increase for longer unlock periods.

Figure 10 shows the effects of unlocks under this initial constant correction control by comparing the accuracy and precision of the zero controller during a period of steady lock (same data as Figure 8) and during a period where there were multiple unlocks. The accuracy is high for both periods as the temperature error distribution is centered around 0. However, the precision is decreased when there are multiple unlocks, as the PDF flattens and the standard deviation increases to 22 mK. This accuracy is outside the north arm tolerance to maintain the 1% finesse asymmetry requirement, and will be exacerbated for longer unlocks.

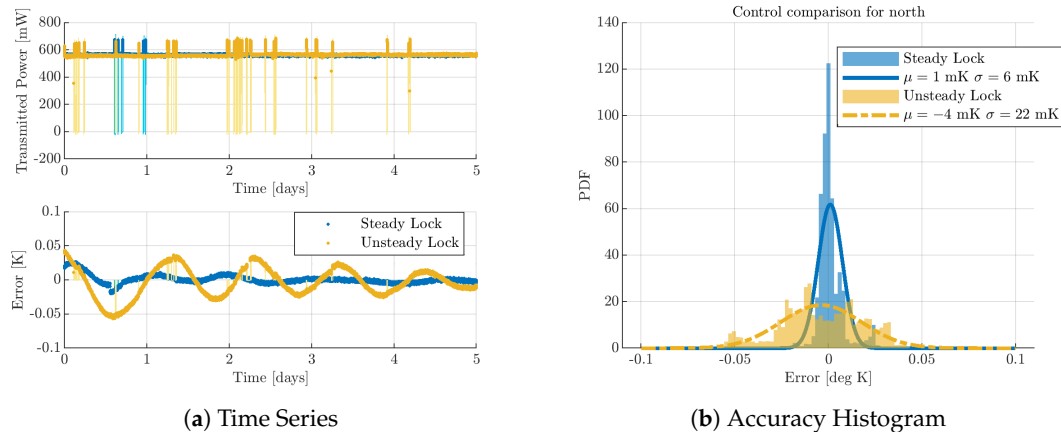

(**a**) Time Series                    (**b**) Accuracy Histogram

**Figure 10.** Comparison of the zero controller accuracy and precision for a period of steady lock and a period of multiple unlocks. The decreased precision during the unsteady period shows the need to develop a more sophisticated control for the unlock condition.

A new unlock condition has been designed to make the absence of the laser heat transfer unknown to the etalon control by using the laser plant identified in Figure 9. When the ITF unlocks, the response of the laser plant to a step input equal to the laser power is used to generate a temperature offset. The offset is subtracted from the set temperature so that the etalon control follows the laser cooling and no step is observed by the control. When the ITF relocks, the step response of the laser plant is used again to return the offset back to zero. When the the offset is within a certain tolerance of zero, it is reset so that the set temperature returns to nominal.

The performances of the two unlock controls are shown in Figure 11, with the new control shown in the top plots and the initial control in the bottom. In Figure 11a,b, the ITF unlocks around the 10 h mark. At this point the offset is shown to increase to match the mirror cooling. The "set–actual" temperature curve is also seen to increase, indicating that without the new unlock control, the error would sharply increase. Instead, the error signal with the new unlock control remains smooth, without the presence of a step. Figure 11b shows that the actuator correction signals remain smooth and predicable during the unlocked periods, indicating that the control is not overcompensating for the laser heating. Note, the data in Figure 11a,b were smoothed to better display the behavior of the down condition control. The data in Figure 11c,d were not smoothed due to the steps in the control when the ITF unlocks and relocks.

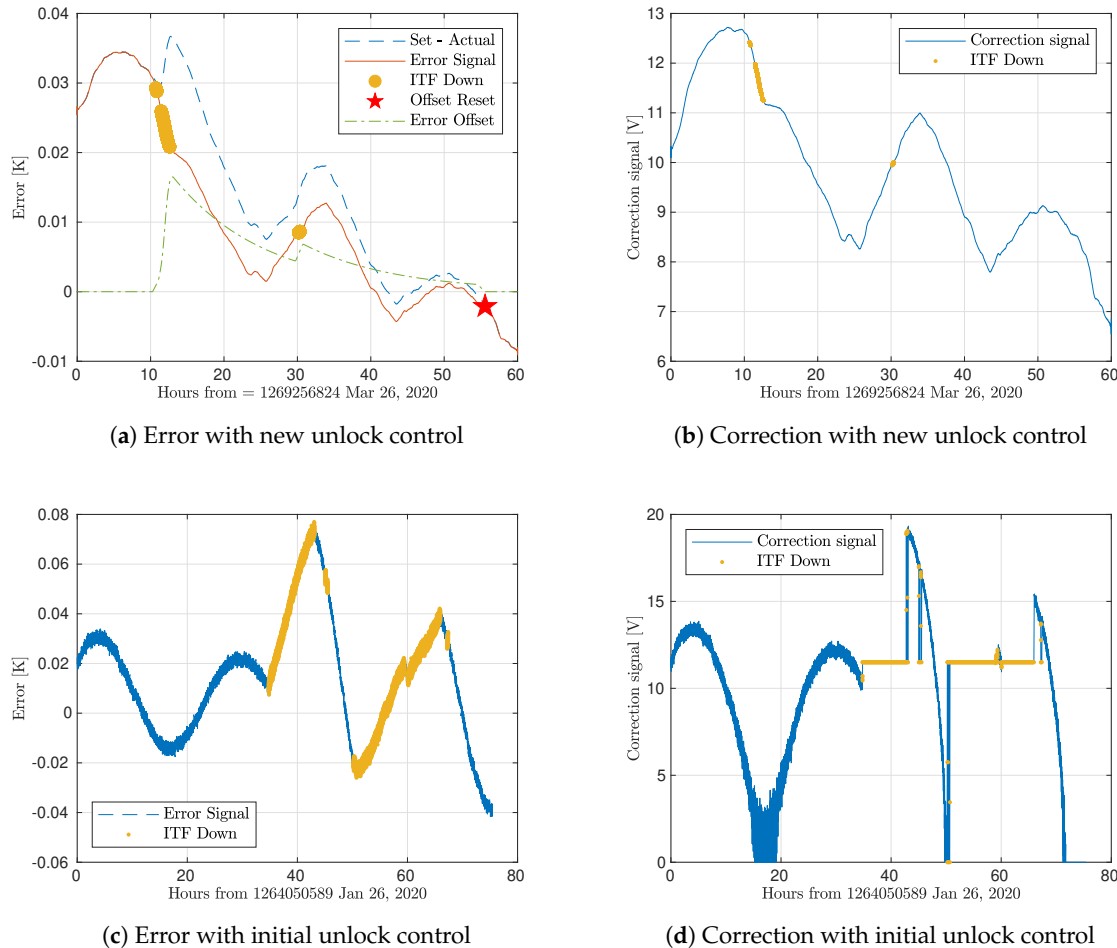

(**a**) Error with new unlock control　　　　　　　　　　　　(**b**) Correction with new unlock control

(**c**) Error with initial unlock control　　　　　　　　　　　(**d**) Correction with initial unlock control

**Figure 11.** Comparison of etalon control performance during an unlock with and without the new unlock control. With the unlock condition, (**a**) shows an offset, calculated from the laser plant, is subtracted from the set temperature to match mirror cooling so that the error is consistently driven to zero. Additionally, (**b**) shows that the correction signal remains smooth during the unlock and relock periods. Without the unlock condition, (**c**) shows that the error increases during unlock due to cooling in the laser absence. When the ITF relocks, the increased error is seen as a step input which causes an overshoot in the correction signal shown in (**d**), particularly as the mirror heats due to the laser.

Figure 11c,d shows the behavior of the control with the initial unlock condition. Up until the unlock at 35 hours, the error and the corrections were smoothly varying to the set temperature. When the ITF unlocks, the error increases sharply as the mirror cools, and the corrections go to the constant value. After the ITF relocks, the increased error is seen as a step for the control which responds with a large correction signal. The combination of this large correction and the laser heating quickly overshoots the set temperature, which is evident by the steep approach to the set temperature. With the new unlock condition, these overshoots are avoided and the error is reliably driven to zero.

Unfortunately, the unlock condition was implemented a few days before the end of the O3b run and no long term data are available for statistics.

## 4. Discussion

Mirror temperature control was implemented in Advanced Virgo to vary the intra-mirror etalon cavity and balance the arm cavity finesse for optimal performance. A two-pole system step response matches the measured plant step response with a high coefficient of determination, where the high frequency pole is identified as heat transfer from the actuator to the surrounding tower and the low

frequency pole is the subsequent radiation heat transfer from the tower to the mirror. The developed zero controller filter achieves 6 mK accuracy with a 11.2 hour response time to 0.67 etalon fringe step input—a notable improvement from the 30 mK accuracy of the previous Virgo+ control. These findings confirm that the etalon effect is a viable feature with which to achieve symmetry between interferometer arms, particularly to supplement manufacturing tolerance capabilities. Finally, a control methodology was developed compensate the temperature error during interferometer unlock periods by adding an offset to the error signal equal to the mirror temperature step response of laser cooling.

The Advanced Virgo detector manufacturing can provide excellent finesse asymmetries (less than 0.5%) [30]. However, the implementation of etalon can further reduce the finesse asymmetry down to 0.06% [31,32] when the optimal working point is set; see Figure 2. Moreover, the etalon provides flexibility to adapt to unexpected degradation of the finesse asymmetry. Furthermore, the choice for the parallel faces needed for etalon simplifies the optical design of the central area in that wedges are no longer needed.

In theory, there is still a large margin to improve the etalon control which makes us confident to achieve a better equalization of the finesse asymmetry with respect the manufacturing. These methodologies and control filters must be continuously developed as new data are generated. Unfortunately, the premature ending of the O3b run limited the amount of data available for analysis. Future implementations of the control should incorporate the modern control state-space representation. This realization will require the calibration of the uncontrolled surface temperatures in order to fully define the heat transfer system. In this way, arbitrary inputs from seasonal temperature variations may be accounted for by the control system.

**Author Contributions:** Conceptualization, J.B., M.M., J.C.D., A.A., and P.R.; methodology, J.B. and M.M.; software, J.B., A.A., and A.M.; formal analysis, J.B. and A.A.; investigation, J.B. and M.M.; resources, A.M. and V.D.; data curation, J.B.; writing–original draft preparation, J.B.; writing—review and editing, J.B., M.M., and A.A.; visualization, J.B.; supervision, M.M.; project administration, M.M. All authors have read and agreed to the published version of the manuscript.

**Funding:** This research received no external funding.

**Acknowledgments:** The authors gratefully acknowledge contributions from Bas Swinkels.

**Conflicts of Interest:** The authors declare no conflict of interest.

## Abbreviations

The following abbreviations are used in this manuscript:

| | |
|---|---|
| AdV | Advanced Virgo |
| ITF | Interferometer |
| AR | Anti-Reflective (Mirror Coating) |
| HR | Highly Reflective (Mirror Coating) |
| OPL | Optical Path Length |
| WCS | Worst Case Scenario |
| TCS | Temperature Compensation System |
| BNS | Binary Neutron Star |
| LTI | Linear Time Invariant (Control System) |
| SI | Simple Integrator |
| UGF | Unity Gain Frequency |
| PDF | Probability Density Function |

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
