# Peer review of "Temperature Control for an Intra-Mirror Etalon in Interferometric Gravitational Wave Detector Fabry–Perot Cavities"

_galaxies, doi:10.3390/galaxies8040080_

Round 1

Reviewer 1 Report

This paper describes the implementation of the temperature control for the input mirrors of the arm cavities of the Advanced Virgo gravitational wave detector. This control is essential to tune the optical transmission of the input mirrors and guarantee a finesse symmetry between the two arms.

The paper is correctly written and the control loop design and performances explained in details. It could be published but after the revisions mentioned below.

In the title: Pérot → Perot (no accent)

L1 gravity waves → gravitational waves (and in the title)

L2 missing a space in this line

L18, “responsible”, a little bit too negative, rephrase here.

Eq 3,4 L_m is the optical length of the substrate, why it is ‘(n-1)’ and not ‘n’ in the equations ?

Eq 5, the dn/dT value used is different from the text above

L50, I would simply say “with the current transmission and AR coating of AdV, the finesse could be tuned by 2.6%”

L58, is it really a sine function or a Airy function ?

P4, section 2.1: I would recommend to remove this whole section, it is overly long and with mistakes and too much simplification for a conclusion rather simple: the temperature evolution has 2 time constants. People will accept it, one time constant for the mirror and one for the heated tower (like say in the conclusion).

For example:

- The thermometer is on the heating ring, so in theory there is also the time constant of the heating ring

- eq 14, the extension of T^4_m – T^4_i is not correct, it should have no T_m in R_r otherwise the following calculations are wrong.

- eq 17, it looks like the only heat exchange by radiation is between the tower and the mirror. The mirror is only very small compared to the tower. What is seeing by the tower is mainly the super-attenuator and that the dominant source of heat exchange. Yes the mirror see mainly the surrounding tower but it is wrong to say that the tower see mainly the mirror.

L180, what is the definition of the “goodness” of the fit ? it could be better to give confidence interval.

L190, what is Acl ?

L196, add plant in front of pole frequency

Figure 6 (left): add vertical line for the plant pole frequencies

Figure 6 (right): for this calculation, is it assumed a cooling and heating for the actuator ? as it is possible to heat but there is no actuator for cooling around the tower.

Figure 7: the calculation of the time constant does not into account the overshoot of the zero controller as we can see from the graph the blue curve is still going down. The calculation should be the same as in figure 6 right to be coherent.

L260 “clipped by a threshold”→ “offset by a constant value” if I understand correctly

Figure 9: top, so we can not see the day or night variation on the temperature ?

Figure 10: why the time constant is different for the heating and cooling ? The difference in the 2 cases is the heating term in equation 23, so should you expect the same time constant ?

L311: I do not think this statement is correct, do you achieve better finesse symmetry in the arms compared to the case of no etalon (like in LIGO) ?

Author Response

Thank you for your comments. The authors believe that the implemented changes greatly improved the quality of the manuscript.

Please see our responses to your comments in the attached PDF.

Reviewer 2 Report

See attached file

Author Response

(The authors gave the same response as above.)

Reviewer 3 Report

The paper "Temperature Control for Intra-Mirror Etalon in Interferometric Gravity Wave Detector Fabry-Perot Cavities" presents results from the Advanced Virgo GW detector, where differences in the finesse values of the two arm cavities can be compensated by exploiting an etalon in the input mirrors, i.e. a resonant field buildup between HR and AR coating. This etalon can be fine-tuned to resonance or anti-resonance by changing the temperature of the mirror itself. The authors demonstrate here their findings for the temperature control of this etalon. The results are of high quality, scientifically sound and certainly relevant for the community, therefore I can recommend the paper for publication in Galaxies.

Overall, I find the presentation a bit on the wordy side and it could be streamlined, however it is readable and understandable also in the current form. I would suggest that the authors start in Section 2 with Figure 4 and significantly reduce the introduction text of Section 2 and/or move it into 2.1 and 2.2. Then Figure 3 should appear in section 2.2. That way, the physical problem would be introduced first, before going into the abstract modelling with the block diagram. Also, in the current form, there are several parameters in Fig3 that are only explained four pages later.

In Figure 2, the "dT" values in the legend are not obvious. I assume they correspond to (half) the difference between the intersection of the red and blue curve with the WCS lines? Maybe this could be made clear by showing these regions with vertical lines.

Some small comments:

l. 31: proposed -> implemented? Also it is written here that the <1% finesse imbalance was outside manufacturing capabilities, however with the values given below Fig. 2, it seems that the HR sides would have fulfilled this and resulted in <0.2% imbalance. Was this a coincidence?

l.137: wording - saying that the laser as a heat source is particularly relevant when the ITF is not locked seems strange, as then this heat source is gone... how can it be relevant then? From later in the manuscript it becomes clear that it is the disappearance of this heat source that is relevant.

Fig. 9 seems to contain a lot of (vector) points which at least on my device led to significant delay in displaying it. Consider converting to pixel and/or reduce number of datapoints.

Author Response

(The authors gave the same response as above.)

Round 2

Reviewer 1 Report

Thanks a lot to the authors for the correction of the manuscript.

Looking at this new version, I still have few comments which could improve the paper.

Q1. Yes, please remove the accent to Perot as his family name has no accent (whatever said the English wikipedia page)

Q4. Thanks for the source of the equation 3 and looking at the origin of the formula, it confirms that it may not be correct to be applied for the Virgo etalon. Andrade et al. derived the formula for the OPL in transmission. Here, it is the OPL for one way in the substrate we are looking for.

OPL(T) = n(T) L(T) = n(T) (1 + \alpha T) L

Q8, eq11 now, R_r could not be assumed constant when it has T_m inside… otherwise we arrived to the conclusion that T_m is constant. I know it is a convenient shortcut but that is incorrect.

The approximation should be:

T_m^4 – T_i ^4 = (T_m – T_i) * (4 T_i^3)

Q18, L311, in the conclusion, for completeness it should be added the origin of the finesse asymetry for Kagra: the 2 input mirrors were not coated at the same time. The case is different for LIGO and Virgo. Also talking about KAGRA due to cryogenic mirrors, the etalon effect may not be operated (much smaller dn/dT and the mirror can not be heated).

Author Response

Thank you again for your thorough review.
